# N-terminal Backbone Pairing Shifts in CCL5-^12^AAA^14^ Dimer Interface: Structural Significance of the FAY Sequence

**DOI:** 10.3390/ijms21051689

**Published:** 2020-03-01

**Authors:** Jin-Ye Li, Yi-Chen Chen, Yi-Zong Lee, Chun-Hsiang Huang, Shih-Che Sue

**Affiliations:** 1Institute of Bioinformatics and Structural Biology, National Tsing Hua University, Hsinchu 30013, Taiwan; ebangirl@hotmail.com (J.-Y.L.); ali17172002@gmail.com (Y.-C.C.); s942645@go.thu.edu.tw (Y.-Z.L.); 2Instrument Center, National Tsing Hua University, Hsinchu 30013, Taiwan; 3Protein Diffraction Group, Experimental Facility Division, National Synchrotron Radiation Research Center, Hsinchu 30076, Taiwan; huang.conann@nsrrc.org.tw; 4Department of Life Science, National Tsing Hua University, Hsinchu 30013, Taiwan

**Keywords:** chemokine, aggregation, polymerization, dimer, sulfate

## Abstract

CC-type chemokine ligand 5 (CCL5) has been known to regulate immune responses by mediating the chemotaxis of leukocytes. Depending on the environment, CCL5 forms different orders of oligomers to interact with targets and create functional diversity. A recent CCL5 trimer structure revealed that the N-terminal conversed F12-A13-Y14 (^12^FAY^14^) sequence is involved in CCL5 aggregation. The CCL5-^12^AAA^14^ mutant with two mutations had a deficiency in the formation of high-order oligomers. In the study, we clarify the respective roles of F12 and Y14 through NMR analysis and structural determination of the CCL5-^12^AAA^14^ mutant where F12 is involved in the dimer assembly and Y14 is involved in aggregation. The CCL5-^12^AAA^14^ structure contains a unique dimer packing. The backbone pairing shifts for one-residue in the N-terminal interface, when compared to the native CCL5 dimer. This difference creates a new structural orientation and leads to the conclusion that F12 confines the native CCL5 dimer configuration. Without F12 anchoring in the position, the interfacial backbone pairing is permitted to slide. Structural plasticity occurs in the N-terminal interaction. This is the first case to report this structural rearrangement through mutagenesis. The study provides a new idea for chemokine engineering and complements the understanding of CCL5 oligomerization and the role of the ^12^FAY^14^ sequence.

## 1. Introduction

CC-type chemokine ligand 5 (CCL5) also known as RANTES (regulated on activation, normal T cell expressed and secreted) is expressed and produced by many types of cells including platelets, macrophages, eosinophils, fibroblasts, and epithelial cells [1]. The molecule participates in a wide range of inflammatory disorders and pathologies [2,3,4]. 

Many chemokines show similar features to cooperatively execute their functions in a concentration-dependent manner for binding to G protein-coupled receptors (GPCRs) as monomers at low concentrations and interact with glycosaminoglycans (GAGs) as oligomer formations at high concentrations [5,6,7,8,9]. CCL5 has a similar concentration-dependent manner, but a more complicated regulatory mechanism [10,11,12,13,14]. The aggregation property correlates with CCL5 chemotaxis activity as disaggregated mutants lose their inflammatory properties, such as T cell activation [15,16]. There are at least three phases for CCL5 oligomerization and aggregation that depend on pH, concentration, and temperature [14]. Under acidic conditions, CCL5 adopts an equilibrium of monomers and dimers [17], while higher ionic strength promotes the association of CCL5 into a dimer. The CCL5 dimer starts to oligomerize when the pH increases to 5.0. The oligomerization process becomes significant as pH approaches neutral. At pH 7.0, massive aggregation can be observed if the concentration is higher than µM.

The first structure of CCL5 (PDB codes 1RTN and 1HRJ) was determined as a dimer by using the NMR method at pH 3.7 [18,19]. The structure represents a typical CC-type chemokine dimer configuration where the interface occurs between the N-terminal regions: a short antiparallel β-sheet pairing by the sequences of T8-P9-C10 (^8^TPC^10^) from the two units. To gain insight into how CCL5 molecules oligomerize, many studies have investigated the mechanism. Residues E26 and E66 were first identified to play roles in the CCL5 self-association where the mutation at residue E26 or E66 resulted in disaggregated conformation, which was a tetramer or dimer, respectively [16]. The E66S mutation was effective and in particular, created a stable CCL5 dimer under a wide range of pH levels and concentrations [15,16,17]. In addition, two oligomeric structures have been reported. The first case is a tetrameric model (PDB code 2L9H) built by the constraints from NMR, small-angle X-ray scattering (SAXS) and mass spectrometry [13]. The tetramer model was built at pH 4.5 and can be assembled into an extended oligomer. The interface for polymerization is composed of residues in the first β-strand (β1) and at the end of the C-terminal α-helix (α1) and contains several hydrophobic and electrostatic interactions. A crucial salt bridge is found between K25 and E66. The second structure is a CCL5 (4–68) hexamer (PDB code 5CMD) [20]. The deletion of the first three residues in the N-terminus caused no change for the dimer structure but favors different interactions for the oligomerization. At the contact interface, E66 interacts with R44 and K45, and E26 forms a salt bridge with R47. Subsequently, the CCL5 dimers stack with each other and constitute a rod-shaped filament in the structure. 

Through NMR and turbidity assays, the two oligomers have been suggested to interactively exist in solution [14]. The introduction of a mutation to abolish the interfacial interaction in either model cannot fully eliminate the CCL5 oligomerization. CCL5 alternatively adopts the two types of interactions to form distinct oligomers and polymers. The observations were concluded as an integrative model to coordinate the oligomerization process [14]. However, the regulation and exact corresponding functions in cells are still missing.

The aggregation behavior of CCL5 could be further explained by another determined structure [14]. A strategy to mix native CCL5 and E66S mutant allows for the crystallization of CCL5 in a limited oligomerized formation [21]. Under an optimized ratio, the CCL5 and E66S mixture was crystalized primarily in a trimer formation, in which one native CCL5 molecule conjugated two CCL5-E66S molecules (PDB 6AEZ). In the structure, a new interface was observed between the native CCL5 (chain B) and CCL5-E66S (chain C). There are two backbone hydrogen bonds between the respective residues, Y14 and Y14′. The interactions link the N-terminal regions of these two molecules. Additional hydrogen bonds are also detected between F12 and A16′. The substitution at Y14 by Ala introduced disaggregated property. Thus, the new interface has been concluded to be involved in CCL5 aggregation and precipitation [14].

By multiple sequence alignments of CCL5s from different animals, CCL5s conversely compose the sequence of F12-A13-Y14 (^12^FAY^14^). This implies the structural and biological significance of the ^12^FAY^14^ sequence [14]. In addition, previous studies have testified the mutation of F12 in CCL5 and the mutant caused a 5000-fold reduction in the receptor binding affinity and a >90% reduction in calcium flux, demonstrating that F12 is significantly connected with the receptor binding [22]. The equivalent importance is also found in other homologs, such as CCL4 (MIP-1β) [23]. 

We studied a CCL5 variant, CCL5-^12^AAA^14^ with two mutations at positions 12 and 14. The mutant using the A12-A13-A14 (^12^AAA^14^) sequence to replace the original ^12^FAY^14^ sequence resulted in a less assembled CCL5 variant [14]. To realize the detailed interaction contributed by the ^12^FAY^14^ sequence, it is necessary to study the specific roles of F12 and Y14. Here, we aim to investigate how the ^12^FAY^14^ sequence acts in CCL5 aggregation and why CCL5-^12^AAA^14^ contains less aggregated property. Using the combination of NMR and X-ray crystallography, we characterized the structure, dynamics, aggregation property, and sulfate binding of CCL5-^12^AAA^14^. We expect that the study will provide a better understanding of the regulation of chemokine oligomerization.

## 2. Results

### 2.1. CCL5-^12^AAA^14^ Mutant

Oligomerization and aggregation are significant characteristics of CCL5. Previous studies have proved that the molecular association of CCL5 is related to pH, concentration, and temperature. Through NMR ^1^H, ^15^N-heteronuclear single quantum coherence spectroscopy (HSQC), we verified the pH-dependent property of CCL5 (Figure 1). At pH 3.2, CCL5 existed in a monomer-dimer equilibrium and the dimer formation was dominant. As the pH increased to 4.0, the CCL5 resonances became uniform and matched the dimer formation. When the pH reached 5.0, the intensities of the resonances significantly decreased. Notably, the sample is still transparent with no floccule or precipitate. We concluded that CCL5 starts to form large and soluble oligomers and only a very low content of protein was still in the dimeric formation. This phenomenon was maintained until pH 6.0. When the pH was higher than 6.0, a cloudy appearance was observed in the NMR sample. All NMR resonances became invisible at pH 7.0, indicating that CCL5 was in the phase of aggregation. 

As mentioned, a CCL5 trimer formation was identified that two CCL5-E66S molecules were conjugated with one wild-type CCL5 molecule. In the interface, there are two hydrogen bonds formed between residues Y14 and Y14′ and F12 contributes to the partial hydrophobic interaction in the dimer interface. The CCL5 variant, CCL5-^12^AAA^14^ with two Ala substitutions at residues F12 and Y14, resulted in the formation of a less aggregated mutant. At an acidic pH, the ^1^H, ^15^N-HSQC spectrum of CCL5-^12^AAA^14^ exhibited more than one set of resonances with unequal intensities (Figure 1). The feature remained until pH was titrated to 6.0. When the pH was higher than 6.0, the resonances exhibited decreased intensities, indicating oligomerization was occurring. However, unlike the native CCL5, CCL5-^12^AAA^14^ remained a transparent sample and some resonances were still visible. The two mutations of F12A and Y14A reduced the CCL5 oligomerization propensity and eliminated the precipitation property. 

### 2.2. Single Mutations of F12A and Y14A

To investigate the respective roles of residues F12 and Y14, two CCL5 variants of CCL5-F12A and CCL5-Y14A were constructed. The Ala substitutions were expected to determine the side chain hydrophobic effects derived from the F12 and Y14 residues. The ^1^H, ^15^N-HSQC spectra of CCL5-F12A collected at pH 3.2 displayed two sets of resonances, corresponding to the monomer and dimer formations (Figure 1). Compared to the native CCL5, the monomer population became more evident. With the pH increased to 5.0, the intensities of the resonances became weaker but still observable in the HSQC spectrum, indicating the formation of a few oligomers. When the pH reached 6.0, the aggregation appeared in the form of a vaporous sedimentation. Overall, we concluded that the replacement at position 12 promoted CCL5 dimer dissociation. This effect reduced the oligomerization tendency, but the aggregation property was still maintained when the pH reached neutral. 

Regarding Y14A, its ^1^H, ^15^N-HSQC spectrum in the acidic condition of pH 3.2, contained more than one set of resonances, corresponding with multiple conformations (Figure 1). When the pH was titrated to 5.0, almost no signals were observed indicating severe oligomerization. Surprisingly, at the condition of pH 7.0, the protein sample had great transparency without aggregation and precipitation. The mutation preserved the oligomerization property but the precipitation property was lost. The Ala substitution at position 14 converted CCL5 to become deficient in precipitation. Residue Y14 is mainly responsible for CCL5 precipitation.

### 2.3. Sulfate Stabilized the CCL5-^12^AAA^14^ Dimer

The complexity of the CCL5-^12^AAA^14^ HSQC spectrum represents a combination behavior of F12A and Y14A, which is less oligomerized and non-precipitated (Figure 1). In acidic conditions, the spectrum reflected the result of multiple conformations in solution. To determine the CCL5-^12^AAA^14^ structure, we tested buffer conditions and found that the presence of a high concentration of sulfate ions unified the CCL5-^12^AAA^14^ conformation. A single set of resonances was detected in the presence of 180 mM ammonium sulfate at pH 3.2 (Figure 2). In the HSQC, sulfate stabilized the major formation and eliminated the other minor forms. Therefore, we were able to crystalize the sulfate-stabilized CCL5-^12^AAA^14^.

### 2.4. Crystal Structure of CCL5-^12^AAA^14^

Crystals of CCL5-^12^AAA^14^ grew in the reservoir solution containing 0.2 M magnesium acetate tetrahydrate, pH 7.9 and 20% PEG3350. The crystals belong to the trigonal space group P3_1_21 with unit-cell parameters *a* = *b* = 48.3 Å and *c* = 60.4 Å and a solvent content of 54%. One CCL5-^12^AAA^14^ molecule appeared in the asymmetric unit that was calculated automatically by CCP4 program suite [24]. For molecular replacement, we used chain A in the native CCL5 dimer structure (PDB code 1EQT) as a template. The structure of CCL5-^12^AAA^14^ monomer was refined to 2.55 Å. The first five N-terminal residues were not modeled due to the lack of interpretable electron density. The structure of the CCL5-^12^AAA^14^ monomer matched the generalized view of a typical chemokine tertiary structure, consisting of an extended N-terminal portion (N–P20), a short 3_10_ helix (R21–H23), a triple-stranded antiparallel β-sheet (β1: I24–F28, β2: V39–T43, and β3: N46–A51) and a C-terminal α-helix (α1: K56–E66). There are two disulfide bridges formed between four cysteine residues, C10–C34 and C11–C50. The Ramachandran plot showed that 98.4% of the residues were found in the most favored and additional allowed regions and 1.6% in the generously allowed region. The statistics of the diffraction data are listed in Table 1. The structures with relatively high root mean square derivations (RMSDs) of bond lengths and angles might reflect the imperfection of the crystal packing, matching the high value of B factors and dynamic features of the monomer/dimer equilibrium in solution. 

Notably, although one CCL5-^12^AAA^14^ molecule existed in one asymmetric unit, it packed as a dimer with two-fold symmetry (Figure 3). In the interface of the CCL5-^12^AAA^14^ dimer, backbone hydrogen bonds were formed between C10 (chain A) and C10′ (chain B) (Figure 3A), T7 and C50′, and T8 and Q48′ (Figure 3B). These hydrogen bonds stabilized the interface constituted by the N-terminal portions and the β3 element. 

### 2.5. N-terminal Backbone Pairing of CCL5-^12^AAA^14^ Dimer

Compared to the monomer unit in the native CCL5 dimer structure, the CCL5-^12^AAA^14^ monomer showed an extremely similar structure. The superposition revealed similarity with RMSDs ranging from 0.370 to 0.447 Å (Figure 4, left). However, there exists a significant difference in the dimer interface. In the native CCL5 dimer, the dimeric interface is a short β-sheet, constituted by ^8^TPC^10^ sequences from the two units (Figure 4B). In CCL5-^12^AAA^14^, the backbone interaction was centralized at C10 (Figure 4A). There is a one-residue shifting in the N-terminal backbone pairing region, which changes the relative orientation of the two monomers. When one of the monomer units of the CCL5-^12^AAA^14^ dimer overlaps on the monomer unit of the native CCL5 dimer, there exists a significant orientation difference between the second monomers (Figure 4C). The overall length and size of the CCL5-^12^AAA^14^ dimer becomes shorter and more compact, while the distance becomes closer between the two α1 elements. The CCL5-^12^AAA^14^ dimer possesses an approximate globular shape and the wild-type CCL5 was elongated. The interface analyzed by the PDBePISA server [25] showed that the CCL5-^12^AAA^14^ dimer has a smaller interfacial area (552 Å^2^) than that of the wild-type CCL5 dimer (732 Å^2^).

### 2.6. NMR Secondary Structural Prediction

To prove the relevance of the dimer observed in X-ray crystallography, we studied CCL5-^12^AAA^14^ by NMR. The backbone assignments were identified using a ^15^N, ^13^C-labeled CCL5-^12^AAA^14^ sample (Figure 5A). The resonances of S1, P18, L19, K25, K33, K45, and E54–R59 were missing, probably due to the line broadening effect, explained by local dynamics with an intermediate-exchange regime in NMR. The NMR backbone chemical shifts reflect protein backbone orientation. The determined ^13^Cα and ^13^Cβ chemical shifts were used to predict the secondary structure of CCL5-^12^AAA^14^. Here, the chemical shift variation, Δδ, is defined as the difference between the observed value and the predefined value from random coil that is Δδ = δ_observed_ − δ_random-coil_. We employed the variations derived from ^13^Cα and ^13^Cβ to predicate its secondary structure by using the parameter, (ΔδCα − ΔδCβ). The derived value of each residue was further normalized with the frontal and posterior residues by the relationship of Δδ_i_ = (Δδ_i-1_ + Δδ_i_ + Δδ_i+1_)/3. A negative value represents a β-strand, and a positive value represents an α-helix. The resulting prediction of CCL5-^12^AAA^14^ well reflected its characteristic secondary structure, including an extended N-terminal portion, followed by three β-strands and an α-helix (Figure 5B). The result reinforces the agreement with the newly determined crystal structure. The same prediction for wild-type CCL5 was plotted and compared. We observed a major difference that occurred at the N-terminal residue 5-12 (Figure 5B, red arrow). Specifically, there is a less β-strand tendency at C10, C11, and A12 of CCL5-^12^AAA^14^. The result matches the dimer model with hydrogen bonds between C10 and C10′, instead of the ^8^TPC^10^-^8′^TPC^10′^ pairing (Figure 4). The reduced β-strand tendency explains the remaining interaction between C10–C10′. Some other minor differences were also found at the residues S5 to T7 and V49 to C50. 

### 2.7. NMR Backbone Dynamics

Protein backbone dynamics correlate with structure in solution, which also provide information on the molecular size. We investigated the backbone dynamics of CCL5-^12^AAA^14^ to differentiate whether CCL5-^12^AAA^14^ adopts a dimer formation in solution. The NMR backbone dynamics experiments were performed to extract R_1_, R_2_ and heteronuclear ^1^H-^15^N NOE information (Figure 6). We compared the results in the absence and presence of sulfate. Since the sulfate stabilized the major conformation of CCL5-^12^AAA^14^ (Figure 2), the backbone assignments were readily transferred back to the condition without sulfate, based on a series of sulfate titration experiments. The sulfate stabilized CCL5-^12^AAA^14^ had averaged R_1_, R_2_, and NOE values of 1.26 s^−1^, 11.47 s^−1^ and 0.70 while the corresponding conformation in the absence of sulfate had the values of 1.38 s^−1^, 10.31 s^−1^ and 0.68, respectively. Compared to the reported cases of the CCL5 monomer and dimer [14,26], these R_1_ and R_2_ values match a dimer formation (Table 2) and the R_2_/R_1_ ratios report overall tumbling rates of ~9 ns, which correspond to a dimer. Thus, the sulfate ions did not change the conformation but enhanced the dimer population. The result confirms the CCL5-^12^AAA^14^ dimer formation in solution. Although the distributions of the relaxation parameters were similar in the two conditions, a few differences were noticed. The sulfate-stabilized dimer had slightly greater R_2_ values, indicating CCL5-^12^AAA^14^ interacting sulfate and with a slower tumbling rate. Additional R_2_ decreases occurred in some residues, such as V39 and N52. The residues exhibited a possible sulfate-induced conformation exchange. Meanwhile, some reduced NOE values were noticed in the residues located near the dimer interface, including D6, A13, and V39.

### 2.8. The Interface of the CCL5-^12^AAA^14^ Dimer

We compared the NH chemical shifts between the wild-type CCL5 and CCL5-^12^AAA^14^ to characterize the structural differences (Figure 7). The chemical shift differences were calculated by the equation of Δδ_NH_ = ([Δδ_H_^2^ + (Δδ_N_/5)^2^]/2)^1/2^; therein, Δδ_NH_ represents the combined chemical shift difference of an amide proton (H) and an amide (N). We noticed the residues of S5 to T8 and C11 to R17 with a significant Δδ_NH_, in which residues T8 and A12 had the highest perturbations (>1 ppm). As mentioned, the secondary structure of CCL5-^12^AAA^14^ is consistent with that of the wild-type CCL5. The chemical shift differences might not reflect the secondary structural differences. Therefore, the difference reflected the new interface in CCL5-^12^AAA^14^. In wild-type CCL5, the amide proton of T8 formed a hydrogen bond with C10′. However, the T8 amide proton showed no interaction in the CCL5-^12^AAA^14^ dimer structure. The large Δδ_NH_ value indicates the absence of hydrogen bonds in T8. Position 12, with a very significant difference, could be due to the substitution from Phe to Ala, causing the chemical environment change. In addition, the ^5^SDT^7^ sequence was affected by the structural contact with A14′, I15′, A16′, and C50′. Other residues C34, N36, and Q48 were located near the interface of the CCL5-^12^AAA^14^ dimer. When mapping the perturbed residues on dimer structure of CCL5-^12^AAA^14^, the result perfectly validates the new dimer conformation, revealing the relevance of the packing dimer in the X-ray crystal.

### 2.9. Sulfate Binding on CCL5

We observed the effect of ammonium sulfate on stabilizing the CCL5-^12^AAA^14^ dimer. The sulfate-induced chemical shift perturbations could reveal the potent sulfate-binding site as well as the stabilizing mechanism (Figure 8). In the presence of 180 mM ammonium sulfate, a number of residues had chemical shift perturbations (Δδ_NH_). Residues with significant perturbations (>0.1 ppm) included S5, T7, T8, V39, and M67 and moderate perturbations (between 0.05 and 0.1 ppm) included residues S4, C10, A13, R21, A22, I24, F28, G32, T43, R47, E60, I62, N63, and L65 (Figure 8A). Among the perturbed residues, only two basic residues of R21 and R47 might contribute to electrostatic interactions with sulfate. The other perturbated residues are distributed dispersedly on the protein surface (Figure 8B). No specific binding sites could be identified in the analysis. Considering that the positively charged residues align on the two sides of the CCL5-^12^AAA^14^ dimer (Figure 8C), the presence of sulfate ions could partially neutralize the charge-charge repulsive forces and thus enhance the dimer population. However, we mentioned some backbone resonances missing in the HSQC, including the highly basic portion, ^55^KKWVR^59^, located at the 50s-loop and other positive residues of K25, K33, and K45. The presence of sulfate might induce backbone dynamics in the basic residues and cause the line-broadening effect. These basic residues might be involved in sulfate binding.

## 3. Discussion

We have previously determined a special trimer complex where one wide-type CCL5 molecule conjugates with two CCL5-E66S molecules [14]. The structure indicates the involvement of the ^12^FAY^14^ motif in regulating CCL5 aggregation and precipitation. Two hydrogen bonds were observed between Y14 of CCL5-E66S and Y14′ of wild-type CCL5 (Figure 9A). The related mutant, CCL5-^12^AAA^14^, was identified as a low oligomerized and disaggregated variant. The substitutions at residues F12 and Y14 caused a deficiency in the formation of CCL5 oligomer and aggregate. Several experimental approaches including NMR, turbidity assay, X-ray crystallography, and mutagenesis have been used to study the CCL5 oligomerization and aggregation process [14]. We further clarify the roles of the individual residues of F12 and Y14 and the structural differences between the native CCL5 and CCL5-^12^AAA^14^ in this study. 

In the native CCL5, the sidechain of F12 contributes to intermolecular interaction with the residues Y3′ and K33′ (Figure 9B,C) and stabilizes the dimer interface. When F12 was mutated to Ala, the intermolecular interaction was abolished. The mutation creates a higher tendency for dissociation. The NMR HSQC spectrum reported an elevated population of monomer in the CCL5-F12A mutant, where the spectrum contained resonances for both dimer and monomer. Together with the fact that F12 is important for dimer assembly, we suggest the role of F12 in confining the local structure of the dimer interface. Residue Y14 participates in CCL5 aggregation. For the single mutation of Y14A, the oligomerization property was preserved but no aggregation and precipitation were observed at a neutral pH. The Y14–Y14′ interface is involved in forming CCL5 insoluble aggregates. In addition, Y14 in the CCL5 structure provides regional interactions. It is buried in a pocket surrounded by the residues N36, P37, and A38 in the 30s loop, as well as A51, N52, and P53 in the 50s loop (Figure 9B,D). The substitution to Ala with a reduced sidechain moiety created the backbone fluctuation. Therefore, A14 might have less tendency to conjugate with A14′ of another CCL5-^12^AAA^14^ molecule. Overall, F12 contributes to the stabilization of the dimer interface, while Y14 participates in aggregation. The mutant with the two mutations, therefore, combines both properties, exhibiting reduced oligomerization and aggregation properties. The proposed integrative model for CCL5 assembly includes three phases: monomer/dimer, soluble oligomer and insoluble aggregate [14]; CCL5-^12^AAA^14^ influences the first and third phases and becomes less assembled in solution.

The ^12^FAY^14^ sequence is conserved in the CCL5 family among different mammalian species. In addition, F12 and Y14 are also conserved in the same positions in CCL3 (MIP-1α), CCL4 (MIP-1β), CCL14, and CCL26 (eotaxin-3, MIP-4α) (Figure 9E) [16,27,28]. The chemokine ligands all bind CCR5 and possess bioactivity for inhibiting HIV infection [16,27]. Particularly, CCL5 shares high sequence similarities to CCL3 and CCL4. Previous studies proved that F13 (corresponding to F12 in CCL5) participates in not only the dimerization of CCL4 but also the interaction with the receptor CCR5 [23]. The replacement of F13 could disrupt the dimerization of CCL4 and the mutant CCL4-F13A showed almost no activity to CCR5 [23]. The same observation occurred in CCL5 binding with CCR3 and CCR5 [22]. In our study, residue F12 of CCL5 contributed to the dimer stability and regulated the dimer configuration, reconfirming the importance of the residue. 

In the presence of sulfate, the CCL5-^12^AAA^14^ dimer was stabilized. The dimer structure determined by X-ray crystallography contains a novel N-terminal backbone pairing shift. In the wild-type CCL5 dimer, the short N-terminal antiparallel β-sheet was formed between the ^8^TPC^10^ sequences of the two monomers. In CCL5-^12^AAA^14^, the interfacial interaction shifted to residues C10 and C10′, accompanied by other intermolecular interactions such as T7 and Q48′, and T8 and C50′. The novel dimer configuration was confirmed with NMR chemical shift differences and relaxation experiments. As mentioned, the presence of F12 confines the wild-type CCL5 dimer configuration. Without F12 anchoring in the position, CCL5 would only have a loose dimer packing; it possibly causes dimer dissociation and re-association into any energetically favorable configuration. The phenomenon was reflected in the multiple sets of resonances in the CCL5-^12^AAA^14^ HSQC in acidic conditions. The presence of sulfate stabilized the CCL5-^12^AAA^14^ dimer into the configuration with the backbone pairing shift. We can explain the conformation selection by steric hindrance. The native CCL5 cannot adopt the CCL5-^12^AAA^14^ dimer configuration because the sidechains of F12 and F12′ are too close to pack together in the structure. The substitution to Ala eliminates the hindrance, permitting the new configuration. Thus, the result reveals structural plasticity in the chemokine N-terminal packing, suggesting that chemokine assembly could be modulated by the “sliding” of the N-terminal interface. This is the first instance of the report that chemokines could rearrange dimer packing through the N-terminal mutation. This CCL5-^12^AAA^14^ dimer configuration can be found in other chemokines. CXCL4 forms a tetramer that comprises two types of dimers [29]. In the structure, the monomer contacts one neighbor unit with a CXC-type dimer configuration and the other neighbor unit with the CCL5-^12^AAA^14^ dimer-type configuration. The interactions create a stable tetramer in solution [30]. A similar configuration has also been found in the CXCL7 (NAP2) tetramer (PDB 1NAP) [31] and in the CXCL10 (IP-10) M form tetramer (PDB 1O7Y) [32]. This tetramer formation has been suspected to be important for GAG binding. Another notable example is CCL17 (TARC). The CC-type chemokine used the C10-C10′ interaction to constitute the N-terminal interface and form a dimer. The relative orientation between the two CCL17 monomer units is similar to the CCL5-^12^AAA^14^ dimer, where the two C-terminal helices are closer to each other in space [33]. We compared the N-terminal sequences of the CC-type chemokines (Figure 9E). The sequence alignment shows that the ^12^FAY^14^ region could be variable in the family. The feature of residue variability might contribute to distinct backbone pairing in their N-terminal interfaces. 

The sulfate group is conjugated with different molecules and regulates physiological responses such as the diversity of sulfation patterns on GAGs and receptor N-terminal Tyr residues. Multiple sulfate-binding epitopes have been reported for CCL5. The ^44^RKNR^47^ motif at the 40s loop has been defined as the primary site for the GAG interaction [34,35,36]. A hexamer structure of CCL5 (4–68) bound with heparin trisaccharide revealed the role of C-terminal ^55^KKWVR^59^ as another GAG-binding site [20]. In addition, the N-terminal region including the N-loop is involved in medium-sized chondroitin sulfate binding, acting as a minor GAG-binding epitope [37]. The residues R17, H23, and K45 contribute in recognizing the doubly sulfated N-terminal segment of CCR5 [38]. Here, we investigated the interaction between the sulfate ion and CCL5-^12^AAA^14^. In the titration experiment, there were no specific sites demonstrating significant sulfate-induced chemical shift perturbations. The induced perturbations were moderate, hence we concluded the effect of the sulfate ion was mainly to promote the dimer population without changing the conformation. However, we noticed many basic residues missing in the HSQC spectra. This is possibly due to sulfate-induced intermediate time-scale conformational exchange that causes NMR resonances to not be detected by NMR. Among the missing residues, the presence of sulfate ions eliminated the entire resonances in the ^54^EKKWVR^59^ sequence that comprises three positive residues. The effect might not be attributed to a non-specific electrostatic effect. We suspect that ^55^KKWVR^59^ serves as a specific site for sulfate binding, but the affinity might not be strong. Since no convincing electron density can be concluded as sulfate ions in our structure, a more detailed experiment is required to define the interaction of sulfate ions on CCL5.

## 4. Materials and Methods

### 4.1. Protein Preparation

The DNA sequence encoding human CCL5 was inserted into a pET43.1a (+) vector. The DNA sequence encoding residues 1–68 of the mature CCL5 was preceded by a starting Met residue (M0). Point mutations for CCL5-F12A, CCL5-Y14A and CCL5-^12^AAA^14^ were performed by standard procedure and all constructs were confirmed by DNA sequencing. Plasmids for CCL5 and the mutants were transformed into the *E. coli* strain BL21 (DE3) strain for protein expression. The cells were cultured in either LB or M9 minimal medium while isotopically labeled proteins for NMR usage were produced by incorporating ^15^NH_4_Cl and ^13^C-glucose into M9 minimal medium as the nitrogen and carbon sources, respectively. When the cells reached an O.D._600_ of 0.8, protein expression was induced by isopropyl-thio-β-D-1-thiogalactopyranoside (IPTG) for 5 h at 37 °C. The cells were harvested by centrifugation and stored at −20 °C before protein purification. The CCL5s were purified based on the previously reported procedure [21]. Briefly, CCL5 fractions were obtained from cell inclusion bodies. After denaturation by 6 M guanidinium chloride and refolding in a dialyzing buffer containing 0.9 M guanidinium chloride, 5 mM cysteine, 5 mM methionine and 100 mM Tris-HCl (pH 8.0), the refolded CCL5 proteins were purified by reversed-phase HPLC using a C18 column. The purities were verified by mass spectrometry. 

### 4.2. X-ray Crystallization 

Initial crystallization conditions were established using the sitting-drop vapor-diffusion method with a set of crystallization screens (Hampton Research, USA). The lyophilized CCL5-^12^AAA^14^ sample with a concentration of 6 mg/mL was prepared in 0.5 μL of 25 mM sodium acetate at pH 4, 150 mM NaCl, and 180 mM ammonium sulfate, and mixed with 0.5 μL of the reservoir solution (0.2 M magnesium acetate tetrahydrate at pH 7.9 and 20% PEG3350). The optimized crystallization condition of CCL5-^12^AAA^14^ was at 10 °C in hanging drops for one week. For X-ray diffraction experiments, crystals were taken stepwise in the well solution supplemented with 10% (*v/v*) glycerol flash-frozen in liquid nitrogen. The diffraction data were collected at Taiwan Photon Source TPS 05A1 at National Synchrotron Radiation Research Center (NSRRC), Hsinchu, Taiwan. X-ray data sets were integrated and scaled using the program HKL2000. The structure of CCL5-^12^AAA^14^ was solved by molecular replacement with the program Phaser MR in the CCP4 program suite [24]. The search model for CCL5-^12^AAA^14^ was the Met-hCCL5 monomer (PDB code 1EQT). A rotation and translation search using the dimer of Met-hCCL5 identified one monomer in the asymmetric unit (Matthew’s coefficient: 2.6). The protein chain models were built using Coot [39]. Computational refinement was finished using Refmac5 in the CCP4 program suite [24]. In order to gain a better structural quality, we have tried to refine the structure with PHENIX. During the refinement, the resolution was improved to 2.45 Å. However, the derived R_work_ and R_free_ factors became too high to be reasonable due to unidentified reasons. To prevent the issue, we excluded weak-intensity and high-resolution reflections and only refined the structure with Refmac5. In conclusion, the strategy improved the crystallographic quality statistics with reasonable R_work_ and R_free_ factors but traded for a reduction in resolution (2.55 Å) and slightly elevated RMSDs for bonds and angles (Table 1). The final crystallographic R_work_ and R_free_ factors are 21.4% and 24.9%, respectively. The Ramachandran plot calculated for the final CCL5-^12^AAA^14^ showed that the conformations of 98.4% of the residues were located within the most favored and additional allowed regions. 

### 4.3. NMR HSQC Spectroscopy for Titration

All NMR samples contained 0.2 mM ^15^N-labeled CCL5s with 25 mM sodium acetate, 150 mM NaCl and 10% D_2_O. For pH titration experiments, the pH ranged from 3.2 to 7.0, regulated by NaOH and HCl. For the ammonium sulfate titration experiments, the concentrations of ammonium sulfate ranged from 0 to 180 mM. The NMR ^1^H, ^15^N-HSQC spectra were recorded on a Bruker Advance 850 MHz spectrometer. The chemical shift perturbations were calibrated using 4,4-dimethyl-4-silapentane-1-sulfonic acid (DSS) as a reference. All NMR data were processed with NMRPipe [40] and analyzed with Sparky [41]. 

### 4.4. NMR Backbone Assignment 

The NMR sample for the backbone assignment contained 0.2 mM ^15^N/^13^C-labeled protein. All backbone assignment experiments were acquired at 298 K and pH 3.2 on a Bruker 600 MHz spectrometer. The assignment experiments including HNCO, HNCOCA, HNCA, HNCOCACB, and HNCACB were collected by nonuniform sampling acquisition [42], processed by NMRPipe [40], and recorded on a Bruker 600 MHz spectrometer. The resonance assignment strategy followed the well-established method by analyzing the triple-resonance experiments [43].

### 4.5. NMR Relaxation Experiments 

The NMR relaxation experiments of CCL5-^12^AAA^14^ were performed at pH 3.2 in the absence or presence of 180 mM ammonium sulfate. The relaxation data were obtained from well-established ^15^N-R_1_, ^15^N-R_2_, ^1^H-^15^N NOE experiments [44] recorded on a Bruker 600 MHz spectrometer at 298 K. For the longitudinal R_1_ measurements, the relaxation delays were 5, 10, 20, 40, 80, 150, 300, 600, 800, and 1000 msec. For the transverse R_2_ measurements, the relaxation delays were 0, 16.96, 33.92, 50.88, 67.84, 84.8, 101.76, 118.72, 135.68, 169.60, 203.52, and 237.44 msec. ^1^H-^15^N heteronuclear NOE experiments were acquired in an interleaved manner with or without ^1^H saturation. The delay between each acquisition was 4 sec.

## 5. Conclusions

In this study, we focused on the CCL5-^12^AAA^14^ mutant and evaluated the role of ^12^FAY^14^ in the modulation of the CCL5 oligomer. The residues of F12 and Y14 are conserved in many CC-type chemokine ligands and we proved them to be important in the CCL5 aggregation and precipitation; F12 contributes to the dimer association and Y14 in the aggregation process. By studying the CCL5-^12^AAA^14^ mutant, a novel CCL5 dimer was reported where the N-terminal backbone pairing shifts for one-residue. This change creates different dimer configurations for CCL5. The finding implies the structural plasticity for the CC-type chemokine dimer interface. For the first case to report the structural rearrangement, the study contributes a new idea for engineering chemokines. The study of the ^12^FAY^14^ region also complements the understanding of CCL5 oligomerization.

## 6. Accession Number

The structure of CCL5-^12^AAA^14^ was deposited in the Protein Data Bank with the accession ID 6LOG. 

## Figures and Tables

**Figure 1 ijms-21-01689-f001:**
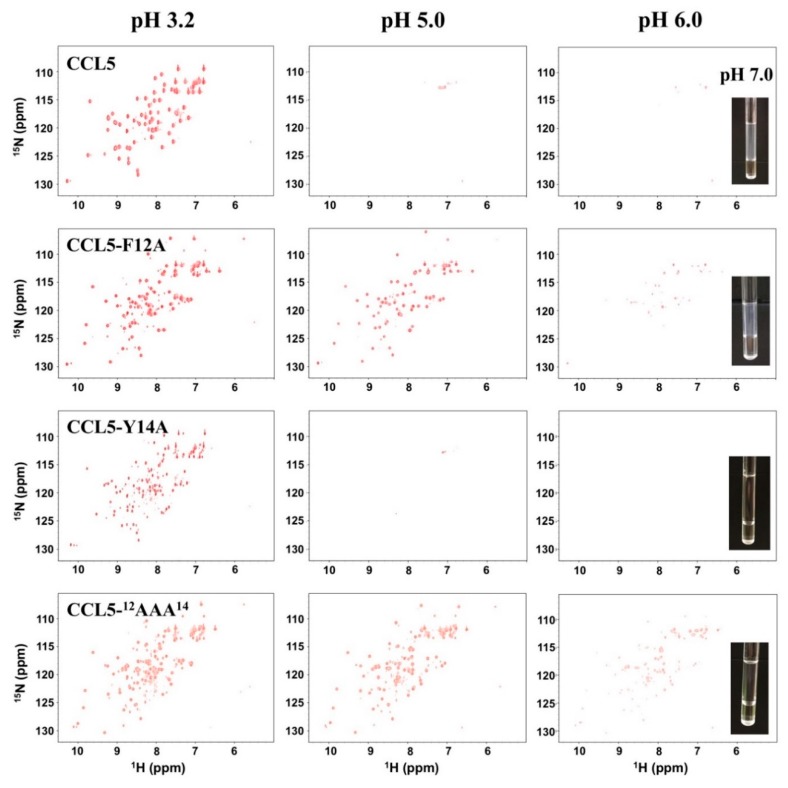
Comparison of the ^1^H,^15^N-HSQC spectra of human CC-type chemokine ligand 5 (CCL5), CCL5-F12A, CCL5-Y14A, and CCL5-^12^AAA^14^ under different pH values. All samples were prepared with a concentration of 0.2 mM in 25 mM sodium acetate and 150 mM sodium chloride. The HSQCs were performed at 298 K. The images of NMR samples at pH 7.0 show different tendencies in protein aggregation and precipitation.

**Figure 2 ijms-21-01689-f002:**
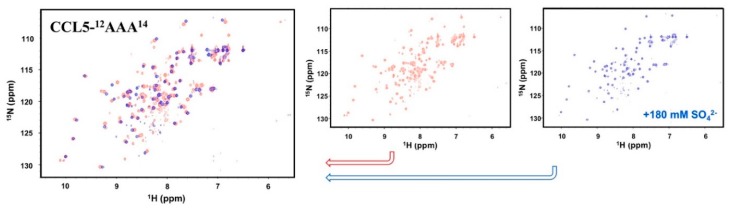
Sulfate titration experiment of CCL5-^12^AAA^14^. ^15^N-CCL5-^12^AAA^14^ was prepared in 25 mM sodium acetate at pH 3.2 with 150 mM of sodium chloride at 298 K. Ammonium sulfate at a final concentration of 180 mM was added into the NMR sample. The corresponding ^1^H,^15^N-HSQC (blue) overlaps with the spectrum before ammonium sulfate titration (red).

**Figure 3 ijms-21-01689-f003:**
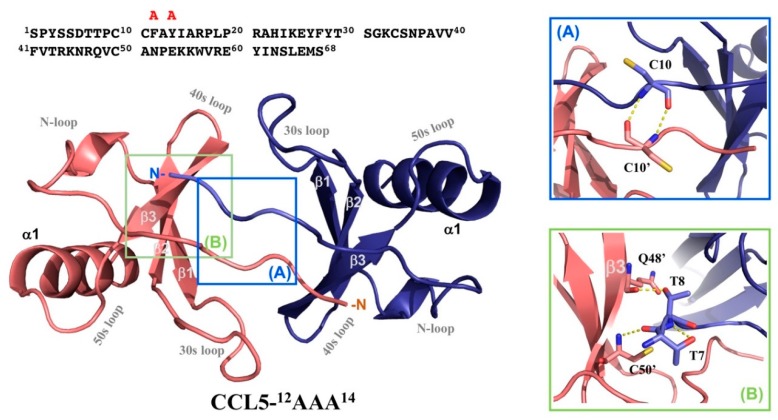
Crystal structure of CCL5-^12^AAA^14^. The structure of CCL5-^12^AAA^14^ solved with a resolution of 2.55 Å adopts typical chemokine fold with three β-sheets and one α-helix. The dimer structure packs with two-fold symmetry. The secondary structural elements and the connecting loops are indicated. The detailed interactions in the interfacial contact region between the two monomers are depicted in (**A**,**B**).

**Figure 4 ijms-21-01689-f004:**
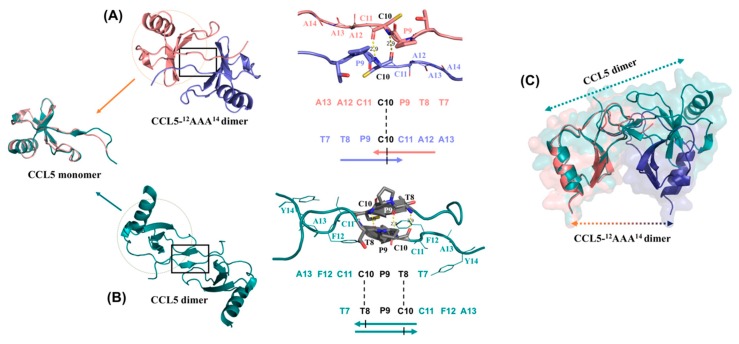
Structural comparison between the CCL5 dimer (PDB code 1EQT) and the CCL5-^12^AAA^14^ dimer. The two dimers share the same monomer structure. The shifts in the N-terminal backbone pairing causes the structural differences. (**A**) The structure of the CCL5-^12^AAA^14^ dimer with a detailed interaction at the dimer interface, centered at residue C10. (**B**) The structure of the CCL5 dimer with a detailed interaction at the dimer interface, centered at the ^8^TPC^10^ sequence. (**C**) The orientation difference between the CCL5-^12^AAA^14^ dimer and the native CCL5 dimer. One of the monomer units of CCL5-^12^AAA^14^ dimer overlaps with the monomer unit of the CCL5 dimer for comparison.

**Figure 5 ijms-21-01689-f005:**
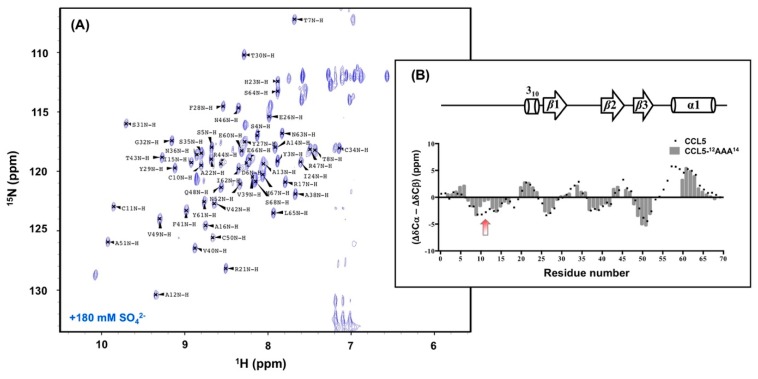
Secondary structure prediction of CCL5-^12^AAA^14^. (**A**) ^1^H,^15^N-HSQC spectrum of CCL5-^12^AAA^14^ with backbone resonance assignments. CCL5-^12^AAA^14^ was prepared in 25 mM sodium acetate at pH 3.2, 150 mM sodium chloride and 180 mM ammonium sulfate. (**B**) The secondary structure of CCL5-^12^AAA^14^ prediction evaluated by the parameter of (ΔδCα − ΔδCβ). The secondary structure determined by X-ray crystallography is referenced on the top. The predication of the native CCL5 is compared and the major difference is noted by a red arrow.

**Figure 6 ijms-21-01689-f006:**
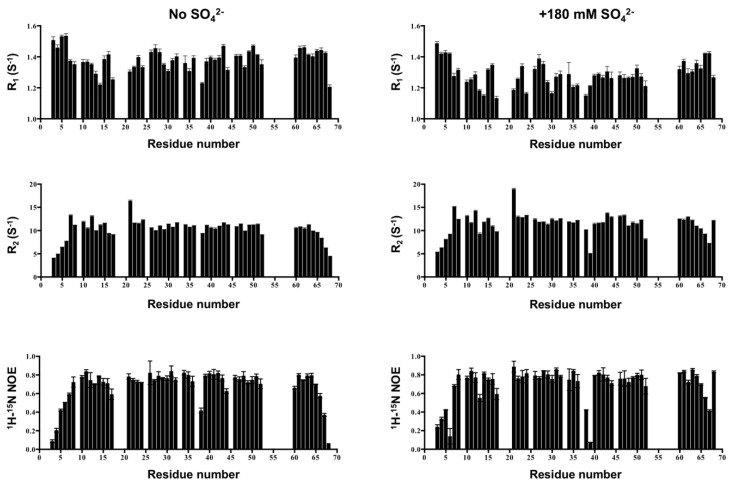
Backbone relaxation data of CCL5-^12^AAA^14^ in the absence or presence of 180 mM ammonium sulfate. ^15^N-R_1_, R_2_ and ^1^H-^15^N NOE (from top to bottom) were obtained for all non-proline residues except for the missing residues (M0, S1, L19, K25, K33, K45, E54, K55, K56, W57, V58, and R59).

**Figure 7 ijms-21-01689-f007:**
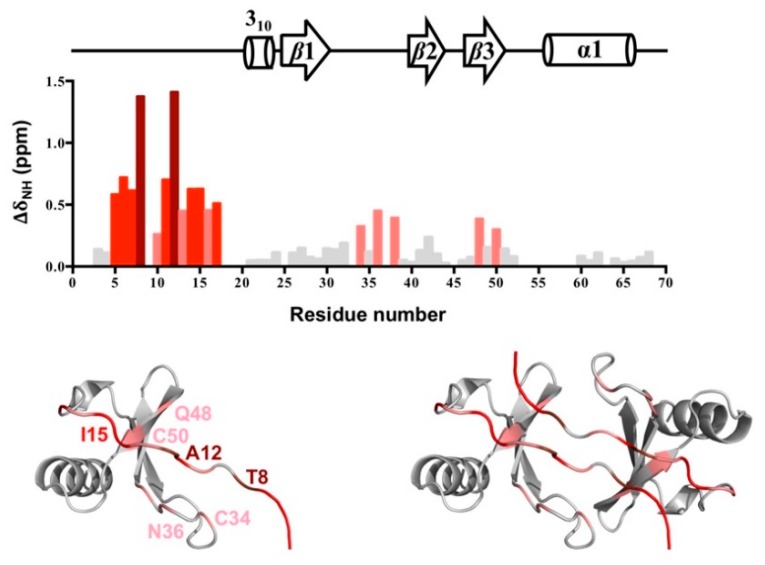
Chemical shift differences between CCL5-^12^AAA^14^ and wild-type CCL5. Chemical shift difference (∆δ_NH_) represents the combined chemical shift difference of the amide and amide proton between the CCL5-^12^AAA^14^ dimer and the wild-type CCL5 dimer. Residues with ∆δ_NH_ > 0.25 ppm are colored by pink, >0.5 ppm by red, and >1.0 ppm by brown. The residues are mapped on the monomer unit and dimer structure of CCL5-^12^AAA^14^ with the same color scheme.

**Figure 8 ijms-21-01689-f008:**
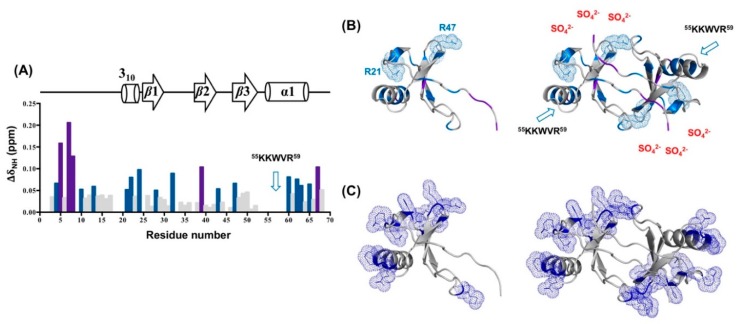
Sulfate-induced perturbations on the CCL5-^12^AAA^14^ dimer. (**A**) Chemical shift perturbations of CCL5-^12^AAA^14^ induced by 180 mM ammonium sulfate. Residues with perturbation >0.05 ppm are colored by blue and >0.1 ppm by purple. (**B**) The perturbated residues are mapped on the monomer unit and dimer structure of CCL5-^12^AAA^14^. Two perturbed basic residues, R21 and R47, and the ^55^KKWVR^59^ sequence are especially noted. (**C**) For comparison, the positively charged residues of CCL5-^12^AAA^14^ are indicated by dots on the monomer unit and dimer structure.

**Figure 9 ijms-21-01689-f009:**
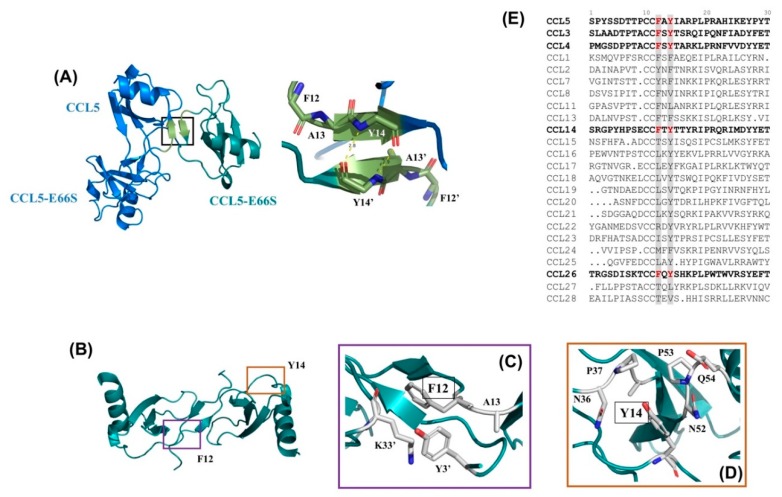
Structure of the CCL5 trimer formation and the interactions of F12 and Y14. (**A**) Overall structure of the CCL5 trimer structure with two CCL5-E66S molecules in complex with one native CCL5 molecule (PDB code 6AEZ). The interface between CCL5-E66S and CCL5 responsible for CCL5 aggregation. (**B**) The locations of F12 and Y14 on the CCL5 dimer (PDBtable code 1EQT). (**C**) A close-up view of the residues near F12. Residue F12 provides intermolecular interactions to stabilize dimer assembly. (**D**) A close-up view of the residues near Y14. Residue Y14 provides hydrophobic interactions with neighboring residues. (**E**) Sequence alignment of the CC-type chemokines for the first 30 residues.

**Table 1 ijms-21-01689-t001:** Data collection and refinement statistics for the CCL5-^12^AAA^14^ crystals.

Data Collection	
Molecule	CCL5-^12^AAA^14^
Beamline	NSRRC TPS 05A1
Wavelength (Å)	0.99984
Space Group	*P*3_1_21
Cell Dimensions	
*a*, *b*, *c* (Å)	48.3, 48.3, 60.4
α, β, γ (^o^)	90, 90, 120
Resolution (Å) ^a^	25.0–2.55 (2.64–2.55)
*R* _merge_ ^b^	0.041 (0.359)
*R_meas_* ^c^	0.049 (0.319)
*R_pim_* ^d^	0.026 (0.212)
CC_1/2_ ^e^	0.978 (0.906)
*I*/*σI*	31.3 (2.2)
Completeness (%)	99.4 (100.0)
Total no. Unique Reflections	2866 (269)
Multiplicity	3.4 (3.4)
*Refinement*	
Resolution (Å)	18.88–2.55 (2.61–2.55)
Unique Reflections	2749 (103)
*R*_work_ (%)	21.4 (32.3)
*R*_free_ (%)	24.9 (44.7)
No. Atoms	
Protein	501
Water	7
*B*-Factors	
Protein	83.06
Water	80.47
RMSD	
Bond Lengths (Å)	0.011
Angles (°)	2.23
Ramachandran	
Most Favored (%)	93.5
Additional Allowed (%)	4.9
Generously Allowed (%)	1.6
Disallowed (%)	0.0
Clash Score	9.02
PDB Accession	6LOG

^a^ Values in parentheses are for the outermost resolution shells. ^b^ R_merge_ = $\sum_{hkl}\sum_{i\;}\left|Ii\left(hkl\right)-\langle I\left(hkl\right)\rangle \right|/\sum_{hkl}\sum_{i}Ii\left(hkl\right)$, where Ii(hkl) is the ith measurement. ^c^ R_meas_ = $\sum_{hkl}{\left[\frac{N}{N-1}\right]}^{1/2}\sum_{i}\left|Ii\left(hkl\right)-\langle I\left(hkl\right)\rangle \right|/\sum_{hkl}\sum_{i}Ii\left(hkl\right)$, where Ii(hkl) is the ith measurement and N is the redundancy of each unique reflection hkl. ^d^ R_pim_ = $\sum_{hkl}{\left[\frac{1}{N-1}\right]}^{1/2}\sum_{i}\left|Ii\left(hkl\right)-<I\left(hkl\right)\right|/\sum_{hkl}\sum_{i}Ii\left(hkl\right)$, where Ii(hkl) is the ith measurement and N is the redundancy of each unique reflection hkl. ^e^ CC_1/2_ is the correlation coefficient between two randomly chosen half data sets.

**Table 2 ijms-21-01689-t002:** NMR dynamic parameters of CCL5s.

	pH	Oligomeric State	R_1_ (s^−1^)	R_2_ (s^−1^)	NOE
Mouse CCL5 *^a^*	3.8	Dimer	1.27 ± 0.05	11.21 ± 0.43	0.69 ± 0.03
5P12-E66S CCL5 *^b^*	3.2	Monomer	2.26 ± 0.13	6.08 ± 0.78	0.71 ± 0.04
CCL5-^12^AAA^14^	3.2	Dimer	1.38 ± 0.013	10.31 ± 0.073	0.68 ± 0.027
CCL5-^12^AAA^14^ (with SO_4_^2-^)	3.2	Dimer	1.26 ± 0.015	11.47 ± 0.097	0.70 ± 0.032

*^a^* Numbers adopted from Reference [14]. *^b^* Numbers adopted from Reference [26].

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
