# Peer review of "N-terminal Backbone Pairing Shifts in CCL5-12AAA14 Dimer Interface: Structural Significance of the FAY Sequence"

_ijms, 2020, doi:10.3390/ijms21051689_

Round 1
Reviewer 1 Report
Review of Sue et al., MS # ijms-705306
The authors have studied the aggregation of chemokine CCL5 and the importance of residues F12 and Y14 in CCL5 aggregation, using H-N HSQC NMR as the main probe. I am not a biophysicist or protein chemist (my experience is with small-molecule NMR), so the following comments are confined to the spectroscopy rather than the biophysics.
This is not a methodological paper: previously-established methods are applied to the current problem. There is no leading reference to the NMR method: readers are expected to understand it. If the NMR data mean what I think, the results are well-grounded; but to so state, I must take the method on faith. I request methodological literature be cited, so that non-specialist readers can follow the logic.
Author Response
We added more methodological literatures (references 40 to 44) for NMR backbone assignment strategy and dynamics study to allow reader following with. However, the methods are really routinely used in protein NMR study, we will not descript too much details in the study.
Reviewer 2 Report
The authors describe a structural and biophysical characterization of a mutant form of the protein CCL5 and the resulting affect on oligomerization. This is a rigorous study with potential broad interest in protein engineering of cytokine ligands, however the manuscript has several shortcomings. First, it requires English language improvement and secondly most figures are too small or poor quality to be visualized by the reader. Furthermore, a few of the conclusions and the significance of these conclusions is over stated. After addressing these concerns it should be acceptable for publication.
Line 47: “determined as a dimer using NMR method” is incorrect English, please fix The following sentence is also grammatically incorrect/awkward
Line 50 TPC has not been defined, please define
Line 53: disaggregated is not a proper word? Unaggregated is the usual word. This sentence also has grammatical errors
Line 66: solutions is grammatically incorrect
Line 82: define FAY. Probably FAY amino acids but it was not clearly stated
Line 93: poor grammar
LIne 97 and rest of manuscript. Poor English grammar, needs correction in the entire paper
Line 99: What kind of HSQC? It is in the figure legend but not the text
The pH values used for oligomerization experiments are extremely far away from physiological pH where CCL5 ligands act. How do the authors reconcile this difference? Low pH is good for NMR, but not biologically relevant in most cases
It is possible the aggregation is artificial due to the ligand being unstable and precipitating near physiological pH. The biological relevance of the conclusions in this paper are unclear without further explanation.
Figure 1: too small and cannot be read. Either break up the figure or make it a full page figure.
Line 135: Regarding Y14A, its HSQC spectrum contained nonuniform resonances at the acidic condition, pH 3.2, corresponding with multiple conformations (Figure 1)
Again, this sentence is not grammatically correct. From a single 1H-15N HSQC the authors cannot ascribe any dynamic properties. Additional data such as NOEs or T1/T2 are required (and provided later in the paper). Furthermore there is great dispersion indicating stable structure. It is possible that the minor peaks are degradation products and not other conformations. The claims of multiple conformations are unfounded in Figure 1. After Figure 2 and the T1/T2 NOE data, then the authors could claim multiple conformations.
Figure 2: Too small, nothing is visible. Please make a bigger and clearer picture
X-ray data collection statistics: The R/Rfree of the model seem to be incorrect. The highest resolution shells are listed as having a lower R/Rfree than the overall data set. That is not possible.
Figure 3 and Figure 4: The authors should provide data demonstrating that the dimerization is not a crystal packing artifact. The authors use NMR tetchqniues to further validate the dimer differences to published structures, but a technique that directed measures particle size in solution such as Dynamic light scattering, or size-exclusion chromatography should be included. A tumbling time can be measured by NMR in addition to T1/T2s, but these are less accurate than DLS or SEC for dimer vs. monomer etc. conclusions.
Additionally, they should provide the total interface area, as per a program such as PISA.
Line 209” The following resonances of S1, P18, L19, K25, K33, K45, and E54- R59 were missing, probably due to the conformational exchange
What conformational change relative to what? The authors should elaborate. I am guessing their structural differences to the published structure?
Figure 5: Again, the figures are too small. Nothing can be read in the HSQC. The authors should use the TALOS program to calculate dihedral angles instead of their method.
Line 315: Determined previously a trimer? The way it reads the authors found a trimer in this paper.
Line 320: F12 and Y14 are crucial for the CCL5 oligomerization and aggregation process. The authors need to elaborate here because this is clearly not the case as shown in this paper. 12AAA14 forms a dimer, and therefore the side chains are not required. Are they instead requires for the trimer from another paper or subtle conformational differences? They go into this further in the next paragraphs, but it should be made clear up front.
Line 378: Thus, the result reveals structural plasticity in the chemokine N-terminal packing that chemokine assembly could be modulated by “sliding” the N-terminal interface. This study reports the structural rearrangement for the first time in chemokine. However, this CCL5-12AAA14 dimer configuration can be found in other chemokines.
These conclusions are unclear and likely unfounded without further explanation. The triple alanine mutation is artificial, so how are the authors able to claim that it is a general mechanism for cytokines? The authors would need to at the very least include a sequence alignment with other cytokines to show that the 12AAA14 region is a variable motif for this family of proteins. If they cannot show this, this conclusion is not founded. The authors do cite several different structures with different configurations, but nothing about a motif or residue variability.
Line 390: sulfate binding. It is likely that the buffer composition simply stabilized the protein construct. This is very common in structural biology and a widely used tool to find buffer compositions that allow protein crystallization.
Line 453: The R/Rfree here do not match the table
Author Response
Response to reviewer 2:
Thanks the careful reading from the reviewer. The suggestions indeed help us to improve the article. We answered all points raised from the reviewer. In short, we reprocessed the X-ray structure and corrected the values of Rwork/Rfree by reducing the resolution. We improved the writing and also corrected the errors.
Line 47: “determined as a dimer using NMR method” is incorrect English, please fix The following sentence is also grammatically incorrect/awkward
Line 50 TPC has not been defined, please define
Page 3: We rewrote the two sentences and defined the TPC.
“was determined as a dimer by using NMR method” …… “pairing by the sequences of T8-P9-C10 (8TPC10)” from the two units”
Line 53: disaggregated is not a proper word? Unaggregated is the usual word. This sentence also has grammatical errors
Page 3: The word is basically taken from the previously publications (JBC, 1999, 274, 16077; JBC, 1999, 274, 27505; Structure, 2011, 19, 1138 and others). To consistently matching the description, we will keep using “disaggregated”. However, we corrected the sentence.
“where the mutation at residue E26 or E66 resulted in disaggregated conformation, which is tetramer or dimer, respectively”
Line 66: solutions is grammatically incorrect
Page 4: We corrected.
Line 82: define FAY. Probably FAY amino acids but it was not clearly stated
Page 4: We modified it to make it clear.
“compose the sequence of F12-A13-Y14 (12FAY14).”
Line 93: poor grammar
Page 4: We modified the writing.
“We expect that the study providing a better understanding of the regulation of chemokine oligomerization.”
LIne 97 and rest of manuscript. Poor English grammar, needs correction in the entire paper
Page 5: We corrected the writing.
“Oligomerization and aggregation are significant characteristics of CCL5.”
Line 99: What kind of HSQC? It is in the figure legend but not the text
We added the description of “1H, 15N-HSQC spectra” and modified other places as well.
The pH values used for oligomerization experiments are extremely far away from physiological pH where CCL5 ligands act. How do the authors reconcile this difference? Low pH is good for NMR, but not biologically relevant in most cases
We agree with the reviewer. However, as mentioned in the Introduction, the structural understanding of CCL5 almost based on the studies in low pH (preventing to form oligomers in neutral pH). Fortunately, the secondary and tertiary structures are kept the same when we reduced pH. The structural difference between the neutral and acidic conditions only occurs in the quaternary structure. To compare with the previously established result, even 12AAA14 mutant is less aggregated in neutral pH, we still used the same strategy to prevent the problem of aggregation. That’s the reason we performed the analysis under low pH. Nevertheless, we believe the current result could still reflect to the situation of physiological condition, as observed in native CCL5 case.
It is possible the aggregation is artificial due to the ligand being unstable and precipitating near physiological pH. The biological relevance of the conclusions in this paper are unclear without further explanation.
Aggregation is not due to protein being unstable in neutral pH. There are already many literatures discussing the aggregation behavior of CCL5, including our recent publication (Chen, YC and et al., JMB, 2020). However, the reviewer gave us a nice suggestion that we should include the description talking about the biological relevance. We added a sentence in Page 3.
“The aggregation property correlates with CCL5 chemotaxis activity that disaggregated mutants lose the inflammatory properties, such as T cell activation”.
Figure 1: too small and cannot be read. Either break up the figure or make it a full page figure.
Uploaded full-size image and increased resolution. We will suggest to make the figure as a full-page figure.
Line 135: Regarding Y14A, its HSQC spectrum contained nonuniform resonances at the acidic condition, pH 3.2, corresponding with multiple conformations (Figure 1)
Again, this sentence is not grammatically correct. From a single 1H-15N HSQC the authors cannot ascribe any dynamic properties. Additional data such as NOEs or T1/T2 are required (and provided later in the paper). Furthermore there is great dispersion indicating stable structure. It is possible that the minor peaks are degradation products and not other conformations. The claims of multiple conformations are unfounded in Figure 1. After Figure 2 and the T1/T2 NOE data, then the authors could claim multiple conformations.
Thank you for the comment. The original writing indeed caused misleading. We would like to point out the different sets of resonances corresponding with multiple conformations. We have no intention to discuss different dynamics distributed in the same protein, here. In addition, we always checked protein SDS-PAGE before and after NMR experiments. The minor population is not for protein degradation. A supporting evidence is the incorporation of sulfate ion converted the minor population into the major conformation. It cannot happen in a degraded sample. Thus, to make it clearer in the article, we corrected the writing in page 6.
“Regarding Y14A, its 1H, 15N-HSQC spectrum at the acidic condition, pH 3.2, contained more than one set of resonances, corresponding with multiple conformations.”
Figure 2: Too small, nothing is visible. Please make a bigger and clearer picture
Uploaded full-size image and increased resolution.
X-ray data collection statistics: The R/Rfree of the model seem to be incorrect. The highest resolution shells are listed as having a lower R/Rfree than the overall data set. That is not possible.
Regarding the mismatching between Rwork/Rfree and resolution, we noticed the problem but had no clear answer. Under the structural refinement by refmac5, the correlation between Rwork/Rfree and resolution is reasonable but the Rwork/Rfree values were relatively high. The following refinement by PHENIX further reduced the values of Rwork and Rfree but meanwhile, created the mismatching. To solve the problem, one solution to improve the crystallographic quality statistics is by systematically excluding weak intensity and high-resolution reflections (Protein Science, 2015, 24, 661-669). The strategy will trade a reduction of resolution. Thus, we tested processes with different resolutions. In final, the dataset of resolution of 2.55 A showed the most satisfactory result. In the calculation, we got 20.1% (25.6%) and 23.2% (35.2%) for Rwork and Rfree, respectively. The values are acceptable. The new structure shows ignorable difference than the original one (rmsd 0.21A). Thus, the change will not cause any difference for our current conclusion. We updated the PDB coordinate and made essential change in the content and table to response the new structural statistics.
Figure 3 and Figure 4: The authors should provide data demonstrating that the dimerization is not a crystal packing artifact. The authors use NMR tetchqniues to further validate the dimer differences to published structures, but a technique that directed measures particle size in solution such as Dynamic light scattering, or size-exclusion chromatography should be included. A tumbling time can be measured by NMR in addition to T1/T2s, but these are less accurate than DLS or SEC for dimer vs. monomer etc. conclusions.
The comment is general right. We have tested size-exclusion chromatography and ultracentrifugation to validate the molecular size. The result is conclusive but because the shape of the dimer is nor globular, we obtained a molecular size slightly higher than the dimer. We therefore to compare NMR T1 and T2 values with the CCL5s that were already known the oligomeric states. Since the comparison is based on the same molecule, it will prevent the influence derived from molecular shape and give the best estimation. However, we also took the suggestion from the reviewer. We converted T1 and T2 values to molecular tumbling rate. We included the information in the content (page 11).
“these R1 and R2 values match a dimer formation (Table 1) and the R2/R1 ratios report overall tumbling rates of ~9 ns, corresponding a dimer”
Additionally, they should provide the total interface area, as per a program such as PISA.
We included the information in page 9.
“The interface analyzed by PDBePISA server showed that the CCL5-12AAA14 dimer has a smaller interfacial area (552 Å2) than that of the wild-type CCL5 dimer (732 Å2).”
Line 209” The following resonances of S1, P18, L19, K25, K33, K45, and E54- R59 were missing, probably due to the conformational exchange
What conformational change relative to what? The authors should elaborate. I am guessing their structural differences to the published structure?
Thank the reviewer to point out the misleading. The missing resonances should be caused by the line broadening effect of local dynamics within ms time scale. To prevent misleading, we rewrote the sentence (page 10).
“the line broadening effect might be explained by local dynamics with an intermediate-exchange regime in NMR.”
Figure 5: Again, the figures are too small. Nothing can be read in the HSQC. The authors should use the TALOS program to calculate dihedral angles instead of their method.
Uploaded full-size image and increased the resolution.
We have examined the dihedral angles by TALOS program. TALOS is based on statistics. The output result did not report any remarkable difference on the dihedral angles between native CCL5 and 12AAA14 mutant. It matched the result that 12AAA14 mutant has the same monomer structure. Thus, TALOS presented no additional information for the comparison. The backbone chemical shifts, however, reported the difference. As expected, we saw the delicate chemical shift changes occurred in the N-terminal segment. That’s an important evidence to supporting the determined 12AAA14 structure (with N-terminal backbone pairing shift). Thus, we will prefer to keep the current method instead of TALOS analysis.
Line 315: Determined previously a trimer? The way it reads the authors found a trimer in this paper.
Page 15: We corrected the writing to made it clear.
Line 320: F12 and Y14 are crucial for the CCL5 oligomerization and aggregation process. The authors need to elaborate here because this is clearly not the case as shown in this paper. 12AAA14 forms a dimer, and therefore the side chains are not required. Are they instead requires for the trimer from another paper or subtle conformational differences? They go into this further in the next paragraphs, but it should be made clear up front.
In the article, oligomerization and aggregation process will be used to present the states with higher-ordered oligomers than dimer. CCL5-12AAA14 only kept moderate ability in forming dimer, however, less potency to form oligomer. To prevent misunderstanding, we rewrote the sentence in page 14
“The substitutions at residues F12 and Y14 caused deficiency in forming CCL5 oligomer and aggregate”
Line 378: Thus, the result reveals structural plasticity in the chemokine N-terminal packing that chemokine assembly could be modulated by “sliding” the N-terminal interface. This study reports the structural rearrangement for the first time in chemokine. However, this CCL5-12AAA14 dimer configuration can be found in other chemokines.
Page 17: We rewrote the sentence to emphasize.
“This is the first time to report that chemokine could rearrange dimer packing through the N-terminal mutation.”
These conclusions are unclear and likely unfounded without further explanation. The triple alanine mutation is artificial, so how are the authors able to claim that it is a general mechanism for cytokines? The authors would need to at the very least include a sequence alignment with other cytokines to show that the 12AAA14 region is a variable motif for this family of proteins. If they cannot show this, this conclusion is not founded. The authors do cite several different structures with different configurations, but nothing about a motif or residue variability.
Page17: We compared the N-terminal sequences of the CC-type chemokines. The sequence alignment shows that the 12FAY14 region could be variable in the family. Only the chemokines, CCL3, CCL4, CCL14 and CCL26 adopt the same FAY sequence. The feature of residue variability might derive different backbone pairing in their N-terminal interfaces. Thus, we include the alignment in Figure 9E to show the variety.
Line 390: sulfate binding. It is likely that the buffer composition simply stabilized the protein construct. This is very common in structural biology and a widely used tool to find buffer compositions that allow protein crystallization.
It might be true for most cases. However, sulfate binding could be important for CCL5. Based on the current study, we have finished sulfate ion binding on native CCL5 recently. CCL5 contain several weak sulfate binding sites based on our titration experiments. We also noticed sulfotyrosine creating better binding specificity. The understanding is important to correlate with GPCR and GAG binding. Thus, in the current study, we found that the presence of sulfate can stabilize 12AAA14 dimer. Studying the interaction between CCL5 and sulfate ion surely aids the understanding of the binding of GPCR and GAG. Therefore, we would like to include the experiment here.
Line 453: The R/Rfree here do not match the table
We corrected and used the new values.

Round 2
Reviewer 2 Report
The authors have made significant improvements to the manuscript addressing my concerns. However, it is still not acceptable for publication. Many of the figure legends are still unreadable and of low quality. This includes Figure 1, Figure 2, Figure 5, and others with structure figure are for some reason incredibly small and low resolution.
Table 2: 1.6% outliers in the Rhamchandran plot is high. Can the authors explain? The bonds and angle RMSD is also typically too high for a crystal structure, in particular the angles.
The crystal structure statistics are still not acceptable for publication without explanation by the authors.
Author Response
For Reviewer 2:
The authors have made significant improvements to the manuscript addressing my concerns. However, it is still not acceptable for publication. Many of the figure legends are still unreadable and of low quality. This includes Figure 1, Figure 2, Figure 5, and others with structure figure are for some reason incredibly small and low resolution.
Thank you for the suggestions. We modified the figures, including Figure 1, 2, 5, 6, 7 and 8. You will find the improvements and the figures now are with better presentations.
Table 2: 1.6% outliers in the Rhamchandran plot is high. Can the authors explain? The bonds and angle RMSD is also typically too high for a crystal structure, in particular the angles.
The crystal structure statistics are still not acceptable for publication without explanation by the authors.
We looked into the residues that was defined as “Outliers”. It is residue A16. However, the original analysis classified “generously allowed” region and “disallowed” region as “Outliers”. The residue is actually distributed in the “generously allowed” region (phi = -49 degree and psi = -80 degree), not “disallowed” region. To prevent the misleading, we clarified the ramachandran plot in a clearer presentation that contain four regions: “most favored”, “additional allowed”, “generously allowed” and “disallowed”. Then, we have no residue distributed in the disallowed region. In addition, A16 has similar dihedral angel as observed in other structures (such as 1EQT). We made the correction in the page 8.
Regarding the high RMSD of bonds and angles. We believe it caused by the imperfection of the crystal packing. For the small protein, people usually can obtain better structure with high resolution. However, this is not the case. The CCL5-AAA mutant is hard to crystalize. Even the presence the sulfate ions allowed the crystallization. We still only reached the resolution of 2.55 A. We suspect it matches the feature of monomer/dimer equilibrium in solution. Nevertheless, as we discussed in the previous letter, we get a reasonable correlation between Rwork/Rfree and resolution if program refmac5 was used. However, software PHENIX created the mismatching values for Rwork/Rfree. Then, we discarded the refinement by PHENIX. The drawback is to bring the elevated RMSDs for bonds and angles. It is hard to find a compensation in this case. Therefore, we decide to present the reasonable Rwork/Rfree values and allow the higher RMSDs. Here, we try to use different model and remove frames to improve the RMSD. Now, we have slightly better values of 0.011A and 2.23 degree. We hope this is a reasonable explanation for the reviewer. However, we would like to notice that the RMSDs is still within the acceptable range if considering from the statistics (see attached figure in the cover letter). This also caused no harm for the paper. We made the correction in page 8 and 22.
Round 3
Reviewer 2 Report
The authors have addressed my comments and the paper is acceptable for publication. The authors should note clearly in their final manuscript the refinement issues (R/Rfree and bonds/angles) appropriately but briefly in their methods section as described in their response.
Author Response
As suggested by reviewer 2. We are glad to add the following sentences in Materials and Methods section (page 21 and 22).
“In order to get a better structural quality, we have tried to refine the structure by PHENIX. During the refinement, the resolution was improved to 2.45 Å. However, the derived Rwork and Rfree factors became too high to be reasonable due to unidentified reasons. To prevent the issue, we excluded weak-intensity and high-resolution reflections and only refined the structure by Refmac5. In final, the strategy improved the crystallographic quality statistics with reasonable Rwork and Rfree factors, but traded a reduction of resolution (2.55 Å) and slightly elevated RMSDs for bonds and angles (Table 1).”